# Unsupervised Curricula
# for Visual Meta-Reinforcement Learning

**Allan Jabri**$^{\alpha}$  **Kyle Hsu**$^{\beta,\dagger}$  **Benjamin Eysenbach**$^{\gamma}$
**Abhishek Gupta**$^{\alpha}$  **Sergey Levine**$^{\alpha}$  **Chelsea Finn**$^{\delta}$

## Abstract

In principle, meta-reinforcement learning algorithms leverage experience across many tasks to learn fast reinforcement learning (RL) strategies that transfer to similar tasks. However, current meta-RL approaches rely on manually-defined distributions of training tasks, and hand-crafting these task distributions can be challenging and time-consuming. Can "useful" pre-training tasks be discovered in an unsupervised manner? We develop an unsupervised algorithm for inducing an adaptive meta-training task distribution, i.e. an *automatic curriculum*, by modeling unsupervised interaction in a visual environment. The task distribution is scaffolded by a parametric density model of the meta-learner's trajectory distribution. We formulate unsupervised meta-RL as information maximization between a latent task variable and the meta-learner's data distribution, and describe a practical instantiation which alternates between integration of recent experience into the task distribution and meta-learning of the updated tasks. Repeating this procedure leads to iterative reorganization such that the curriculum adapts as the meta-learner's data distribution shifts. In particular, we show how discriminative clustering for visual representation can support trajectory-level task acquisition and exploration in domains with pixel observations, avoiding pitfalls of alternatives. In experiments on vision-based navigation and manipulation domains, we show that the algorithm allows for unsupervised meta-learning that transfers to downstream tasks specified by hand-crafted reward functions and serves as pre-training for more efficient supervised meta-learning of test task distributions.

## 1 Introduction

The discrepancy between animals and learning machines in their capacity to gracefully adapt and generalize is a central issue in artificial intelligence research. The simple nematode *C. elegans* is capable of adapting foraging strategies to varying scenarios [9], while many higher animals are driven to acquire reusable behaviors even without extrinsic task-specific rewards [64, 45]. It is unlikely that we can build machines as adaptive as even the simplest of animals by exhaustively specifying shaped rewards or demonstrations across all possible environments and tasks. This has inspired work in reward-free learning [28], intrinsic motivation [55], multi-task learning [11], meta-learning [50], and continual learning [59].

An important aspect of generalization is the ability to share and transfer ability between related tasks. In reinforcement learning (RL), a common strategy for multi-task learning is conditioning the policy on side-information related to the task. For instance, *contextual* policies [49] are conditioned on a task description (e.g. a *goal*) that is meant to modulate the strategy enacted by the policy. Meta-learning of reinforcement learning (meta-RL) is yet more general as it places the burden of inferring the task on the learner itself, such that task descriptions can take a wider range of forms, the most general being an MDP. In principle, meta-reinforcement learning (meta-RL) requires an agent to distill previous

---

$^{\alpha}$UC Berkeley $^{\beta}$University of Toronto $^{\gamma}$Carnegie Mellon University $^{\delta}$Stanford University
$^{\dagger}$Work done as a visiting student researcher at UC Berkeley.

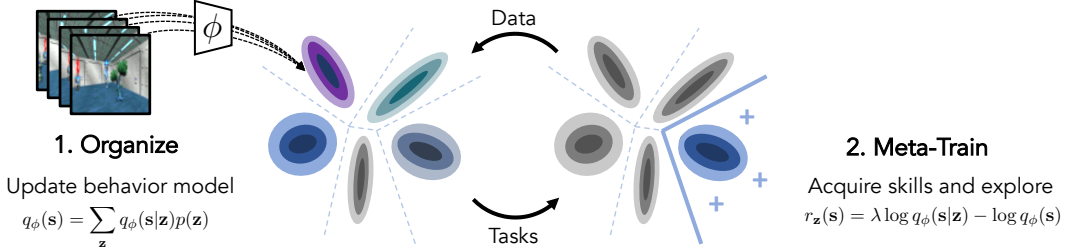

**1. Organize**

Update behavior model

$$q_\phi(\mathbf{s}) = \sum_{\mathbf{z}} q_\phi(\mathbf{s}|\mathbf{z})p(\mathbf{z})$$

Data

Tasks

**2. Meta-Train**

Acquire skills and explore

$$r_{\mathbf{z}}(\mathbf{s}) = \lambda \log q_\phi(\mathbf{s}|\mathbf{z}) - \log q_\phi(\mathbf{s})$$

Figure 1: An illustration of CARML, our approach for unsupervised meta-RL. We choose the behavior model $q_\phi$ to be a Gaussian mixture model in a jointly, discriminatively learned embedding space. An automatic curriculum arises from periodically re-organizing past experience via fitting $q_\phi$ and meta-learning an RL algorithm for performance over tasks specified using reward functions from $q_\phi$.

experience into fast and effective adaptation strategies for new, related tasks. However, the meta-RL framework by itself does not prescribe where this experience should come from; typically, meta-RL algorithms rely on being provided fixed, hand-specified task distributions, which can be tedious to specify for simple behaviors and intractable to design for complex ones [27]. These issues beg the question of whether "useful" task distributions for meta-RL can be generated automatically.

In this work, we seek a procedure through which an agent in an environment with visual observations can automatically acquire useful (i.e. utility maximizing) behaviors, as well as how and when to apply them – in effect allowing for *unsupervised* pre-training in visual environments. Two key aspects of this goal are: 1) learning to operationalize strategies so as to adapt to new tasks, i.e. meta-learning, and 2) unsupervised learning and exploration in the absence of explicitly specified tasks, i.e. skill acquisition *without* supervised reward functions. These aspects interact insofar as the former implicitly relies on a task curriculum, while the latter is most effective when compelled by what the learner can and cannot do. Prior work has offered a pipelined approach for unsupervised meta-RL consisting of unsupervised skill discovery followed by meta-learning of discovered skills, experimenting mainly in environments that expose low-dimensional ground truth state [25]. Yet, the aforementioned relation between skill acquisition and meta-learning suggests that they should not be treated separately.

Here, we argue for closing the loop between skill acquisition and meta-learning in order to induce an *adaptive* task distribution. Such co-adaptation introduces a number of challenges related to the stability of learning and exploration. Most recent unsupervised skill acquisition approaches optimize for the discriminability of induced modes of behavior (i.e. *skills*), typically expressing the discovery problem as a cooperative game between a policy and a learned reward function [24, 16, 1]. However, relying solely on discriminability becomes problematic in environments with high-dimensional (image-based) observation spaces as it results in an issue akin to mode-collapse in the task space. This problem is further complicated in the setting we propose to study, wherein the policy data distribution is that of a meta-learner rather than a contextual policy. We will see that this can be ameliorated by specifying a hybrid discriminative-generative model for parameterizing the task distribution.

The main contribution of this paper is an approach for inducing a task curriculum for unsupervised meta-RL in a manner that scales to domains with pixel observations. Through the lens of information maximization, we frame our unsupervised meta-RL approach as variational expectation-maximization (EM), in which the E-step corresponds to fitting a task distribution to a meta-learner's behavior and the M-step to meta-RL on the current task distribution with reinforcement for both skill acquisition and exploration. For the E-step, we show how deep discriminative clustering allows for trajectory-level representations suitable for learning diverse skills from pixel observations. Through experiments in vision-based navigation and robotic control domains, we demonstrate that the approach i) enables an unsupervised meta-learner to discover and meta-learn skills that transfer to downstream tasks specified by human-provided reward functions, and ii) can serve as pre-training for more efficient supervised meta-reinforcement learning of downstream task distributions.

## 2 Preliminaries: Meta-Reinforcement Learning

*Supervised* meta-RL optimizes an RL algorithm $f_\theta$ for performance on a hand-crafted distribution of tasks $p(\mathcal{T})$, where $f_\theta$ might take the form of an recurrent neural network (RNN) implementing a learning algorithm [13, 61], or a function implementing a gradient-based learning algorithm [18]. Tasks are Markov decision processes (MDPs) $\mathcal{T}_i = (\mathcal{S}, \mathcal{A}, r_i, P, \gamma, \rho, T)$ consisting of state space $\mathcal{S}$,

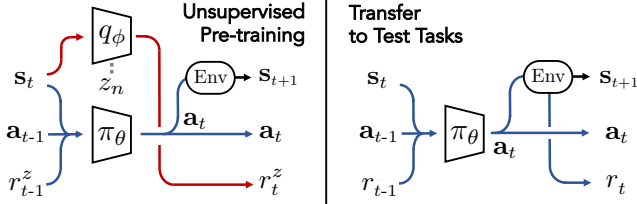

Figure 2: A step for the meta-learner. **(Left) Unsupervised pre-training.** The policy meta-learns self-generated tasks based on the behavior model $q_\phi$. **(Right) Transfer.** Faced with new tasks, the policy transfers acquired meta-learning strategies to maximize unseen reward functions.

action space $\mathcal{A}$, reward function $r_i : \mathcal{S} \times \mathcal{A} \to \mathbb{R}$, probabilistic transition dynamics $P(\mathbf{s}_{t+1}|\mathbf{s}_t, \mathbf{a}_t)$, discount factor $\gamma$, initial state distribution $\rho(\mathbf{s}_1)$, and finite horizon $T$. Often, and in our setting, tasks are assumed to share $\mathcal{S}, \mathcal{A}$. For a given $\mathcal{T} \sim p(\mathcal{T})$, $f_\theta$ learns a policy $\pi_\theta(\mathbf{a}|\mathbf{s}, \mathcal{D}_\mathcal{T})$ conditioned on task-specific experience. Thus, a meta-RL algorithm optimizes $f_\theta$ for expected performance of $\pi_\theta(\mathbf{a}|\mathbf{s}, \mathcal{D}_\mathcal{T})$ over $p(\mathcal{T})$, such that it can generalize to unseen test tasks also sampled from $p(\mathcal{T})$.

For example, RL$^2$ [13, 61] chooses $f_\theta$ to be an RNN with weights $\theta$. For a given task $\mathcal{T}$, $f_\theta$ hones $\pi_\theta(\mathbf{a}|\mathbf{s}, \mathcal{D}_\mathcal{T})$ as it recurrently ingests $\mathcal{D}_\mathcal{T} = (\mathbf{s}_1, \mathbf{a}_1, r(\mathbf{s}_1, \mathbf{a}_1), d_1, \dots )$, the sequence of states, actions, and rewards produced via interaction within the MDP. Crucially, the same task is seen several times, and the hidden state is not reset until the next task. The loss is the negative discounted return obtained by $\pi_\theta$ across episodes of the same task, and $f_\theta$ can be optimized via standard policy gradient methods for RL, backpropagating gradients through time and across episode boundaries.

*Unsupervised* meta-RL aims to break the reliance of the meta-learner on an explicit, upfront specification of $p(\mathcal{T})$. Following Gupta et al. [25], we consider a controlled Markov process (CMP) $\mathcal{C} = (\mathcal{S}, \mathcal{A}, P, \gamma, \rho, T)$, which is an MDP without a reward function. We are interested in the problem of learning an RL algorithm $f_\theta$ via unsupervised interaction within the CMP such that once a reward function $r$ is specified at test-time, $f_\theta$ can be readily applied to the resulting MDP to efficiently maximize the expected discounted return.

Prior work [25] pipelines skill acquisition and meta-learning by pairing an unsupervised RL algorithm DIAYN [16] and a meta-learning algorithm MAML [18]: first, a contextual policy is used to discover skills in the CMP, yielding a finite set of learned reward functions distributed as $p(r)$; then, the CMP is combined with a frozen $p(r)$ to yield $p(\mathcal{T})$, which is fed to MAML to meta-learn $f_\theta$. In the next section, we describe how we can generalize and improve upon this pipelined approach by jointly performing skill acquisition as the meta-learner learns and explores in the environment.

## 3 Curricula for Unsupervised Meta-Reinforcement Learning

Meta-learning is intended to prepare an agent to efficiently solve new tasks related to those seen previously. To this end, the meta-RL agent must balance 1) exploring the environment to infer which task it should solve, and 2) visiting states that maximize reward under the inferred task. The duty of unsupervised meta-RL is thus to present the meta-learner with tasks that allow it to practice task inference and execution, without the need for human-specified task distributions. Ideally, the task distribution should exhibit both structure and diversity. That is, the tasks should be distinguishable and not excessively challenging so that a developing meta-learner can infer and execute the right skill, but, for the sake of generalization, they should also encompass a diverse range of associated stimuli and rewards, including some beyond the current scope of the meta-learner. Our aim is to strike this balance by inducing an adaptive task distribution.

With this motivation, we develop an algorithm for unsupervised meta-reinforcement learning in visual environments that constructs a task distribution without supervision. The task distribution is derived from a latent-variable density model of the meta-learner's cumulative behavior, with exploration based on the density model driving the evolution of the task distribution. As depicted in Figure 1, learning proceeds by alternating between two steps: **organizing experiential data** (i.e., trajectories generated by the meta-learner) by modeling it with a mixture of latent components forming the basis of "skills", and meta-reinforcement learning by **treating these skills as a training task distribution**.

Learning the task distribution in a data-driven manner ensures that tasks are feasible in the environment. While the induced task distribution is in no way guaranteed to align with test task distributions, it may yet require an implicit understanding of structure in the environment. This can indeed be seen from our visualizations in §5, which demonstrate that acquired tasks show useful structure, though in some settings this structure is easier to meta-learn than others. In the following, we formalize our approach, CARML, through the lens of information maximization and describe a concrete instantiation that scales to the vision-based environments considered in §5.

## 3.1 An Overview of CARML

We begin from the principle of information maximization (IM), which has been applied across unsupervised representation learning [4, 3, 41] and reinforcement learning [39, 24] for organization of data involving latent variables. In what follows, we organize data from our policy by maximizing the mutual information (MI) between state trajectories $\boldsymbol{\tau} := (\mathbf{s}_1, \ldots, \mathbf{s}_T)$ and a latent task variable $\mathbf{z}$. This objective provides a principled manner of trading-off structure and diversity: from $I(\boldsymbol{\tau}; \mathbf{z}) := H(\boldsymbol{\tau}) - H(\boldsymbol{\tau}|\mathbf{z})$, we see that $H(\boldsymbol{\tau})$ promotes coverage in policy data space (i.e. *diversity*) while $-H(\boldsymbol{\tau}|\mathbf{z})$ encourages a lack of diversity under each task (i.e. *structure* that eases task inference).

We approach maximizing $I(\boldsymbol{\tau}; \mathbf{z})$ exhibited by the meta-learner $f_\theta$ via variational EM [3], introducing a variational distribution $q_\phi$ that can intuitively be viewed as a task scaffold for the meta-learner. In the E-step, we fit $q_\phi$ to a reservoir of trajectories produced by $f_\theta$, re-organizing the cumulative experience. In turn, $q_\phi$ gives rise to a task distribution $p(\mathcal{T})$: each realization of the latent variable $\mathbf{z}$ induces a reward function $r_{\mathbf{z}}(\mathbf{s})$, which we combine with the CMP $\mathcal{C}_i$ to produce an MDP $\mathcal{T}_i$ (Line 8). In the M-step, $f_\theta$ meta-learns the task distribution $p(\mathcal{T})$. Repeating these steps forms a curriculum in which the task distribution and meta-learner co-adapt: each M-step adapts the meta-learner $f_\theta$ to the updated task distribution, while each E-step updates the task scaffold $q_\phi$ based on the data collected during meta-training. Pseudocode for our method is presented in Algorithm 1.

---

**Algorithm 1:** CARML – Curricula for Automatic Reinforcement of Meta-Learning

---

1: **Require:** $\mathcal{C}$, an MDP without a reward function
2: Initialize $f_\theta$, an RL algorithm parameterized by $\theta$.
3: Initialize $\mathcal{D}$, a reservoir of state trajectories, via a randomly initialized policy.
4: **while** not done **do**
5:     Fit a task-scaffold $q_\phi$ to $\mathcal{D}$, e.g. by using Algorithm 2.             **E-step §3.2**
6:     **for** a desired mixture model-fitting period **do**
7:         Sample a latent task variable $\mathbf{z} \sim q_\phi(\mathbf{z})$.
8:         Define the reward function $r_{\mathbf{z}}(\mathbf{s})$, e.g. by Eq. 8, and a task $\mathcal{T} = \mathcal{C} \cup r_{\mathbf{z}}(\mathbf{s})$.
9:         Apply $f_\theta$ on task $\mathcal{T}$ to obtain a policy $\pi_\theta(\mathbf{a}|\mathbf{s}, \mathcal{D}_\mathcal{T})$ and trajectories $\{\boldsymbol{\tau}_i\}$.
10:        Update $f_\theta$ via a meta-RL algorithm, e.g. RL$^2$ [13].           **M-step §3.3**
11:        Add the new trajectories to the reservoir: $\mathcal{D} \leftarrow \mathcal{D} \cup \{\boldsymbol{\tau}_i\}$.
12: **Return:** a meta-learned RL algorithm $f_\theta$ tailored to $\mathcal{C}$

---

## 3.2 E-Step: Task Acquisition

The purpose of the E-step is to update the task distribution by integrating changes in the meta-learner's data distribution with previous experience, thereby allowing for re-organization of the task scaffold. This data is from the *post-update* policy, meaning that it comes from a policy $\pi_\theta(\mathbf{a}|\mathbf{s}, \mathcal{D}_\mathcal{T})$ conditioned on data collected by the meta-learner for the respective task. In the following, we abuse notation by writing $\pi_\theta(\mathbf{a}|\mathbf{s}, \mathbf{z})$ – conditioning on the latent task variable $\mathbf{z}$ rather than the task experience $\mathcal{D}_\mathcal{T}$.

The general strategy followed by recent approaches for skill discovery based on IM is to lower bound the objective by introducing a variational posterior $q_\phi(\mathbf{z}|\mathbf{s})$ in the form of a classifier. In these approaches, the E-step amounts to updating the classifier to discriminate between data produced by different skills as much as possible. A potential failure mode of such an approach is an issue akin to mode-collapse in the task distribution, wherein the policy drops modes of behavior to favor easily discriminable trajectories, resulting in a lack of diversity in the task distribution and no incentive for exploration; this is especially problematic when considering high-dimensional observations. Instead, here we derive a generative variant, which allows us to account for explicitly capturing modes of behavior (by optimizing for likelihood), as well as a direct mechanism for exploration.

We introduce a variational distribution $q_\phi$, which could be e.g. a (deep) mixture model with discrete $\mathbf{z}$ or a variational autoencoder (VAE) [34] with continuous $\mathbf{z}$, lower-bounding the objective:

$$I(\boldsymbol{\tau}; \mathbf{z}) = -\sum_{\boldsymbol{\tau}} \pi_\theta(\boldsymbol{\tau}) \log \pi_\theta(\boldsymbol{\tau}) + \sum_{\boldsymbol{\tau},\mathbf{z}} \pi_\theta(\boldsymbol{\tau}, \mathbf{z}) \log \pi_\theta(\boldsymbol{\tau}|\mathbf{z}) \tag{1}$$

$$\geq -\sum_{\boldsymbol{\tau}} \pi_\theta(\boldsymbol{\tau}) \log \pi_\theta(\boldsymbol{\tau}) + \sum_{\boldsymbol{\tau},\mathbf{z}} \pi_\theta(\boldsymbol{\tau}|\mathbf{z}) q_\phi(\mathbf{z}) \log q_\phi(\boldsymbol{\tau}|\mathbf{z}) \tag{2}$$

The E-step corresponds to optimizing Eq. 2 with respect to $\phi$, and thus amounts to fitting $q_\phi$ to a reservoir of trajectories $\mathcal{D}$ produced by $\pi_\theta$:

$$\max_\phi \mathbb{E}_{\mathbf{z} \sim q_\phi(\mathbf{z}), \boldsymbol{\tau} \sim \mathcal{D}} \big[ \log q_\phi(\boldsymbol{\tau}|\mathbf{z}) \big] \tag{3}$$

What remains is to determine the form of $q_\phi$. We choose the variational distribution to be a state-level mixture density model $q_\phi(\mathbf{s}, \mathbf{z}) = q_\phi(\mathbf{s}|\mathbf{z})q_\phi(\mathbf{z})$. Despite using a state-level generative model, we can treat $\mathbf{z}$ as a trajectory-level latent by computing the trajectory-level likelihood as the factorized product of state likelihoods (Algorithm 2, Line 4). This is useful for obtaining trajectory-level tasks; in the M-step (§3.3), we map samples from $q_\phi(\mathbf{z})$ to reward functions to define tasks for meta-learning.

---

**Algorithm 2:** Task Acquisition via Discriminative Clustering

1: **Require:** a set of trajectories $\mathcal{D} = \{(\mathbf{s}_1, \ldots, \mathbf{s}_T)\}_{i=1}^{N}$
2: Initialize $(\phi_w, \phi_m)$, encoder and mixture parameters.
3: **while** not converged **do**
4:   Compute $L(\phi_m; \boldsymbol{\tau}, z) = \sum_{\mathbf{s}_t \in \boldsymbol{\tau}} \log q_{\phi_m}(g_{\phi_w}(\mathbf{s}_t)|z)$.
5:   $\phi_m \leftarrow \arg\max_{\phi'_m} \sum_{i=1}^{N} L(\phi'_m; \boldsymbol{\tau}_i, z)$ (via MLE)
6:   $\mathcal{D} := \{(\mathbf{s}, y := \arg\max_k q_{\phi_m}(z = k|g_{\phi_w}(\mathbf{s})))\}$.
7:   $\phi_w \leftarrow \arg\max_{\phi'_w} \sum_{(\mathbf{s},y) \in \mathcal{D}} \log q(y|g_{\phi'_w}(\mathbf{s}))$
8: **Return:** a mixture model $q_\phi(\mathbf{s}, z)$

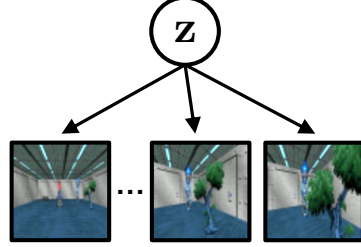

Figure 3: Conditional independence assumption for states along a trajectory.

---

**Modeling Trajectories of Pixel Observations.** While models like the variational autoencoder have been used in related settings [40], a basic issue is that optimizing for reconstruction treats all pixels equally. We, rather, will tolerate *lossy* representations as long as they capture *discriminative* features useful for stimulus-reward association. Drawing inspiration from recent work on unsupervised feature learning by clustering [6, 10], we propose to fit the trajectory-level mixture model via discriminative clustering, striking a balance between discriminative and generative approaches.

We adopt the optimization scheme of DeepCluster [10], which alternates between i) clustering representations to obtain pseudo-labels and ii) updating the representation by supervised learning of pseudo-labels. In particular, we derive a trajectory-level variant (Algorithm 2) by forcing the responsibilities of all observations in a trajectory to be the same (see Appendix A.1 for a derivation), leading to state-level visual representations optimized with trajectory-level supervision.

The conditional independence assumption in Algorithm 2 is a simplification insofar as it discards the order of states in a trajectory. However, if the dynamics exhibit continuity and causality, the visual representation might yet capture temporal structure, since, for example, attaining certain observations might imply certain antecedent subtrajectories. We hypothesize that a state-level model can regulate issues of over-expressive sequence encoders, which have been found to lead to skills with undesirable attention to details in dynamics [1]. As we will see in §5, learning representations under this assumption still allows for learning visual features that capture trajectory-level structure.

### 3.3 M-Step: Meta-Learning

Using the task scaffold updated via the E-step, we meta-learn $f_\theta$ in the M-step so that $\pi_\theta$ can be quickly adapted to tasks drawn from the task scaffold. To define the task distribution, we must specify a form for the reward functions $r_\mathbf{z}(\mathbf{s})$. To allow for state-conditioned Markovian rewards rather than non-Markovian trajectory-level rewards, we lower-bound the trajectory-level MI objective:

$$I(\boldsymbol{\tau}; \mathbf{z}) = \frac{1}{T} \sum_{t=1}^{T} H(\mathbf{z}) - H(\mathbf{z}|\mathbf{s}_1, ..., \mathbf{s}_T) \geq \frac{1}{T} \sum_{t=1}^{T} H(\mathbf{z}) - H(\mathbf{z}|\mathbf{s}_t) \tag{4}$$

$$\geq \mathbb{E}_{\mathbf{z} \sim q_\phi(\mathbf{z}), \mathbf{s} \sim \pi_\theta(\mathbf{s}|\mathbf{z})} \big[ \log q_\phi(\mathbf{s}|\mathbf{z}) - \log \pi_\theta(\mathbf{s}) \big] \tag{5}$$

We would like to optimize the meta-learner under the variational objective in Eq. 5, but optimizing the second term, the policy's state entropy, is in general intractable. Thus, we make the simplifying assumption that the fitted variational marginal distribution matches that of the policy:

$$\max_\theta \mathbb{E}_{\mathbf{z} \sim q_\phi(\mathbf{z}), \mathbf{s} \sim \pi_\theta(\mathbf{s}|\mathbf{z})} \big[ \log q_\phi(\mathbf{s}|\mathbf{z}) - \log q_\phi(\mathbf{s}) \big] \tag{6}$$

$$= \max_\theta I(\pi_\theta(\mathbf{s}); q_\phi(\mathbf{z})) - D_{\mathrm{KL}}(\pi_\theta(\mathbf{s}|\mathbf{z}) \,\|\, q_\phi(\mathbf{s}|\mathbf{z})) + D_{\mathrm{KL}}(\pi_\theta(\mathbf{s}) \,\|\, q_\phi(\mathbf{s}))) \tag{7}$$

Optimizing Eq. 6 amounts to maximizing the reward of $r_\mathbf{z}(\mathbf{s}) = \log q_\phi(\mathbf{s}|\mathbf{z}) - \log q_\phi(\mathbf{s})$. As shown in Eq. 7, this corresponds to information maximization between the policy's state marginal and the latent task variable, along with terms for matching the task-specific policy data distribution to the

corresponding mixture mode and deviating from the mixture's marginal density. We can trade-off between component-matching and exploration by introducing a weighting term $\lambda \in [0, 1]$ into $r_{\mathbf{z}}(\mathbf{s})$:

$$r_{\mathbf{z}}(\mathbf{s}) = \lambda \log q_\phi(\mathbf{s}|\mathbf{z}) - \log q_\phi(\mathbf{s}) \tag{8}$$

$$= (\lambda - 1) \log q_\phi(\mathbf{s}|\mathbf{z}) + \log q_\phi(\mathbf{z}|\mathbf{s}) + C \tag{9}$$

where $C$ is a constant with respect to the optimization of $\theta$. From Eq. 9, we can interpret $\lambda$ as trading off between discriminability of skills and task-specific exploration. Figure 4 shows the effect of tuning $\lambda$ on the structure-diversity trade-off alluded to at the beginning of §3.

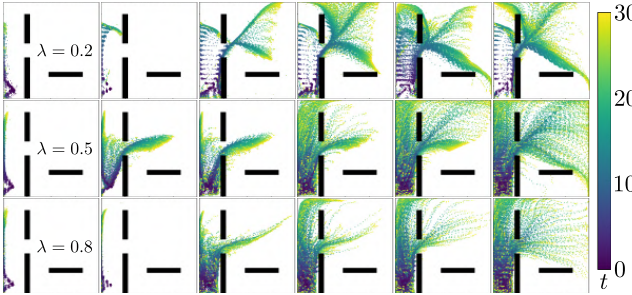

Figure 4: Balancing consistency and exploration with $\lambda$ in a simple 2D maze environment. Each row shows a progression of tasks developed over the course of training. Each box presents the mean reconstructions under a VAE $q_\phi$ (Appendix C) of 2048 trajectories. Varying $\lambda$ of Eq. 8 across rows, we observe that a small $\lambda$ (top) results in aggressive exploration; a large $\lambda$ (bottom) yields relatively conservative behavior; and a moderate $\lambda$ (middle) produces sufficient exploration and a smooth task distribution.

## 4 Related Work

**Unsupervised Reinforcement Learning**. Unsupervised learning in the context of RL is the problem of enabling an agent to learn about its environment and acquire useful behaviors without human-specified reward functions. A large body of prior work has studied exploration and intrinsic motivation objectives [51, 48, 43, 22, 8, 5, 35, 42]. These algorithms do not aim to acquire skills that can be operationalized to solve tasks, but rather try to achieve wide coverage of the state space; our objective (Eq. 8) reduces to pure density-based exploration with $\lambda = 0$. Hence, these algorithms still rely on slow RL [7] in order to adapt to new tasks posed at test-time. Some prior works consider unsupervised pre-training for efficient RL, but these works typically focus on settings in which exploration is not as much of a challenge [63, 17, 14], focus on goal-conditioned policies [44, 40], or have not been shown to scale to high-dimensional visual observation spaces [36, 54]. Perhaps most relevant to our work are unsupervised RL algorithms for learning reward functions via optimizing information-theoretic objectives involving latent skill variables [24, 1, 16, 62]. In particular, with a choice of $\lambda = 1$ in Eq. 9 we recover the information maximization objective used in prior work [1, 16], besides the fact that we simultenously perform meta-learning. The setting of training a contextual policy with a classifier as $q_\phi$ in our proposed framework (see Appendix A.3) provides an interpretation of DIAYN as implicitly doing trajectory-level clustering. Warde-Farley et al. [62] also considers accumulation of tasks, but with a focus on goal-reaching and by maintaining a goal reservoir via heuristics that promote diversity.

**Meta-Learning**. Our work is distinct from above works in that it formulates a meta-learning approach to explicitly train, without supervision, for the ability to adapt to new downstream RL tasks. Prior work [31, 33, 2] has investigated this unsupervised meta-learning setting for image classification; the setting considered herein is complicated by the added challenges of RL-based policy optimization and exploration. Gupta et al. [25] provides an initial exploration of the unsupervised meta-RL problem, proposing a straightforward combination of unsupervised skill acquisition (via DIAYN) followed by MAML [18] with experiments restricted to environments with fully observed, lower-dimensional state. Unlike these works and other meta-RL works [61, 13, 38, 46, 18, 30, 26, 47, 56, 58], we close the loop to jointly perform task acquisition and meta-learning so as to achieve an automatic curriculum to facilitate joint meta-learning and task-level exploration.

**Automatic Curricula**. The idea of automatic curricula has been widely explored both in supervised learning and RL. In supervised learning, interest in automatic curricula is based on the hypothesis that exposure to data in a specific order (i.e. a non-uniform curriculum) may allow for learning harder tasks more efficiently [15, 51, 23]. In RL, an additional challenge is exploration; hence, related work in RL considers the problem of *curriculum generation*, whereby the task distribution is designed to guide exploration towards solving complex tasks [20, 37, 19, 52] or unsupervised pre-training [57, 21]. Our work is driven by similar motivations, though we consider a curriculum in the setting of meta-RL and frame our approach as information maximization.

# 5 Experiments

We experiment in visual navigation and visuomotor control domains to study the following questions:

- What kind of tasks are discovered through our task acquisition process (the E-step)?
- Do these tasks allow for meta-training of strategies that transfer to test tasks?
- Does closing the loop to jointly perform task acquisition and meta-learning bring benefits?
- Does pre-training with CARML accelerate meta-learning of test task distributions?

Videos are available at the project website `https://sites.google.com/view/carml`.

## 5.1 Experimental Setting

The following experimental details are common to the two vision-based environments we consider. Other experimental are explained in more detail in Appendix B.

**Meta-RL.** CARML is agnostic to the meta-RL algorithm used in the M-step. We use the $RL^2$ algorithm [13], which has previously been evaluated on simpler visual meta-RL domains, with a PPO [53] optimizer. Unless otherwise stated, we use four episodes per trial (compared to the two episodes per trial used in [13]), since the settings we consider involve more challenging task inference.

**Baselines.** We compare against: 1) PPO from scratch on each evaluation task, 2) pre-training with random network distillation (RND) [8] for unsupervised exploration, followed by fine-tuning on evaluation tasks, and 3) supervised meta-learning on the test-time task distribution, as an oracle.

**Variants.** We consider variants of our method to ablate the role of design decisions related to task acquisition and joint training: 4) *pipelined* (most similar to [25]) – task acquisition with a contextual policy, followed by meta-RL with $RL^2$; 5) *online discriminator* – task acquisition with a purely discriminative $q_\phi$ (akin to online DIAYN); and 6) *online pretrained-discriminator* – task acquisition with a discriminative $q_\phi$ initialized with visual features trained via Algorithm 2.

## 5.2 Visual Navigation

The first domain we consider is first-person visual navigation in ViZDoom [32], involving a room filled with five different objects (drawn from a set of 50). We consider a setup akin to those featured in [12, 65] (see Figure 3). The true state consists of continuous 2D position and continuous orientation, while observations are egocentric images with limited field of view. Three discrete actions allow for turning right or left, and moving forward. We consider two ways of sampling the CMP $\mathcal{C}$. **Fixed**: fix a set of five objects and positions for both unsupervised meta-training and testing. **Random**: sample five objects and randomly place them (thereby randomizing the state space and dynamics).

**Visualizing the task distribution**. Modeling pixel observations reveals trajectory-level organization in the underlying true state space (Figure 5). Each map portrays trajectories of a mixture component, with position encoded in 2D space and orientation encoded in the jet color-space; an example of interpreting the maps is shown left of the legend. The components of the mixture model reveal structured groups of trajectories: some components correspond to exploration of the space (marked with green border), while others are more strongly directed towards specific areas (blue border). The skill maps of the fixed and random environments are qualitatively different: tasks in the fixed room tend towards interactions with objects or walls, while many of the tasks in the random setting sweep

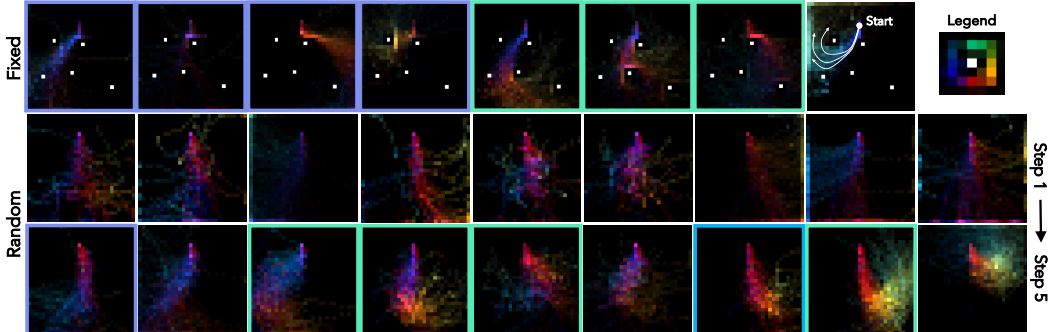

Figure 5: Skill maps for visual navigation. We visualize some of the discovered tasks by projecting trajectories of certain mixture components into the true state space. White dots correspond to fixed objects. The legend indicates orientation as color; on its left is an interpretation of the depicted component. Some tasks seem to correspond to exploration of the space (green border), while others are more directed towards specific areas (blue border). Comparing tasks earlier and later in the curriculum (step 1 to step 5), we find an increase in structure.

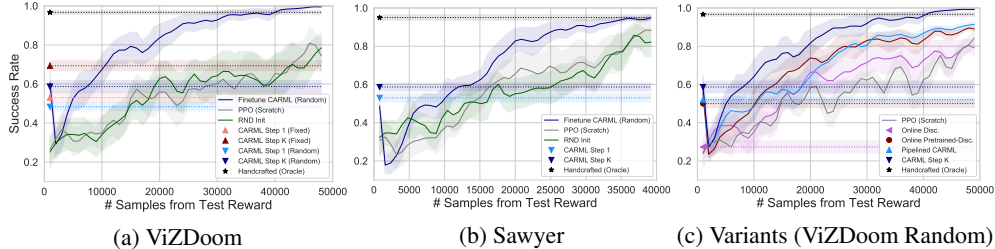

| (a) ViZDoom | (b) Sawyer | (c) Variants (ViZDoom Random) |

Figure 6: CARML enables unsupervised meta-learning of skills that transfer to downstream tasks. Direct transfer curves (marker and dotted line) represent a meta-learner deploying for just 200 time steps at test time. Compared to CARML, PPO and RND Init sample the test reward function orders of magnitude more times to perform similarly on a single task. Finetuning the CARML policy also allows for solving individual tasks with significantly fewer samples. The ablation experiments (c) assess both direct transfer and finetuning for each variant. Compared to variants, the CARML task acquisition procedure results in improved transfer due to mitigation of task mode-collapse and adaptation of the task distribution.

the space in a particular direction. We can also see the evolution of the task distribution at earlier and later stages of Algorithm 1. While initial tasks (produced by a randomly initialized policy) tend to be less structured, we later see refinement of certain tasks as well as the emergence of others as the agent collects new data and acquires strategies for performing existing tasks.

**Do acquired skills transfer to test tasks**? We evaluate how well the CARML task distribution prepares the agent for unseen tasks. For both the fixed and randomized CMP experiments, each test task specifies a dense goal-distance reward for reaching a single object in the environment. In the randomized environment setting, the target objects at test-time are held out from meta-training. The PPO and RND-initialized baseline polices, and the finetuned CARML meta-policy, are trained for a single target (a specific object in a fixed environment), with 100 episodes per PPO policy update.

In Figure 6a, we compare the success rates on test tasks as a function of the number of samples with supervised rewards seen from the environment. Direct transfer performance of meta-learners is shown as points, since in this setting the $RL^2$ agent sees only *four episodes* (200 samples) at test-time, without any parameter updates. We see that direct transfer is significant, achieving up to 71% and 59% success rates on the fixed and randomized settings, respectively. The baselines require over two orders of magnitude more test-time samples to solve a single task at the same level.

While the CARML meta-policy does not consistently solve the test tasks, this is not surprising since no information is assumed about target reward functions during unsupervised meta-learning; inevitable discrepancies between the meta-train and test task distributions will mean that meta-learned strategies *will* be suboptimal for the test tasks. For instance, during testing, the agent sometimes 'stalls' before the target object (once inferred), in order to exploit the inverse distance reward. Nevertheless, we also see that finetuning the CARML meta-policy *trained on random* environments on individual tasks is more sample efficient than learning from scratch. This suggests that deriving reward functions from our mixture model yields useful tasks insofar as they facilitate learning of strategies that transfer.

**Benefit of reorganization**. In Figure 6a, we also compare performance across early and late outer-loop iterations of Algorithm 1, to study the effect of adapting the task distribution (the CARML E-step) by reorganizing tasks and incorporating new data. In both cases, number of outer-loop iterations $K = 5$. Overall, the refinement of the task distribution, which we saw in Figure 5, leads improved to transfer performance. The effect of reorganization is further visualized in the Appendix F.

**Variants**. From Figure 6c, we see that the purely online discriminator variant suffers in direct transfer performance; this is due to the issue of mode-collapse in task distribution, wherein the task distribution lacks diversity. Pretraining the discriminator encoder with Algorithm 2 mitigates mode-collapse to an extent, improving task diversity as the features and task decision boundaries are first fit on a corpus of (randomly collected) trajectories. Finally, while the distribution of tasks eventually discovered by the pipelined variant may be diverse and structured, meta-learning the corresponding tasks from scratch is harder. More detailed analysis and visualization is given in Appendix E.

## 5.3 Visual Robotic Manipulation

To experiment in a domain with different challenges, we consider a simulated Sawyer arm interacting with an object in MuJoCo [60], with end-effector continous control in the 2D plane. The observation is a bottom-up view of a surface supporting an object (Figure 7); the camera is stationary, but the view is no longer egocentric and part of the observation is proprioceptive. The test tasks involve

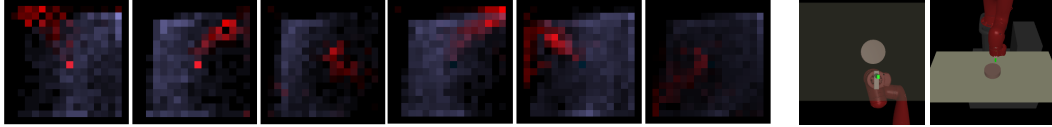

Figure 7: (Left) Skill maps for visuomotor control. Red encodes the true position of the object, and light blue that of the end-effector. Tasks correspond to moving the object to various regions (see Appendix D for more skills maps and analysis). (Right) Observation and third person view from the environment, respectively.

pushing the object to a goal (drawn from the set of reachable states), where the reward function is the negative distance to the goal state. A subset of the skill maps is provided below.

**Do acquired skills directly transfer to test tasks**? In Figure 6b, we evaluate the meta-policy on the test task distribution, comparing against baselines as previously. Despite the increased difficulty of control, our approach allows for meta-learning skills that transfer to the goal distance reward task distribution. We find that transfer is weaker compared to the visual navigation (fixed version): one reason may be that the environment is not as visually rich, resulting in a significant gap between the CARML and the object-centric test task distributions.

### 5.4 CARML as Meta-Pretraining

Another compelling form of transfer is pre-training of an initialization for accelerated supervised meta-RL of target task distributions. In Figure 8, we see that the initialization learned by CARML enables effective supervised meta-RL with significantly fewer samples. To separate the effect of the learning the recurrent meta-policy and the visual representation, we also compare to only initializing the pre-trained encoder. Thus, while

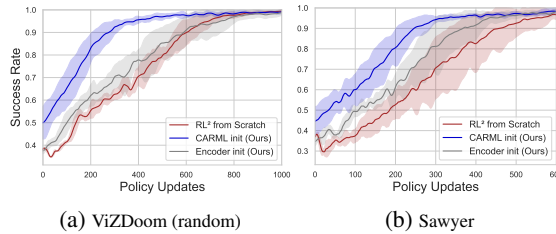

(a) ViZDoom (random)    (b) Sawyer

Figure 8: Finetuning the CARML meta-policy allows for accelerated meta-learning of the target task distribution. Curves reflect error bars across three random seeds.

direct transfer of the meta-policy may not directly result in optimal behavior on test tasks, accelerated learning of the test task distribution suggests that the acquired meta-learning strategies may be useful for learning related task distributions, effectively acting as pre-training procedure for meta-RL.

## 6 Discussion

We proposed a framework for inducing unsupervised, adaptive task distributions for meta-RL that scales to environments with high-dimensional pixel observations. Through experiments in visual navigation and manipulation domains, we showed that this procedure enables unsupervised acquisition of meta-learning strategies that transfer to downstream test task distributions in terms of direct evaluation, more sample-efficient fine-tuning, and more sample-efficient supervised meta-learning. Nevertheless, the following key issues are important to explore in future work.

**Task distribution mismatch**. While our results show that useful structure can be meta-learned in an unsupervised manner, results like the stalling behavior in ViZDoom (see §5.2) suggest that direct transfer of unsupervised meta-learning strategies suffers from a no-free-lunch issue: there will always be a gap between unsupervised and downstream task distributions, and more so with more complex environments. Moreover, the semantics of target tasks may not necessarily align with especially discriminative visual features. This is part of the reason why transfer in the Sawyer domain is less successful. Capturing other forms of structure useful for stimulus-reward association might involve incorporating domain-specific inductive biases into the task-scaffold model. Another way forward is the semi-supervised setting, whereby data-driven bias is incorporated at meta-training time.

**Validation and early stopping**: Since the objective optimized by the proposed method is non-stationary and in no way guaranteed to be correlated with objectives of test tasks, one must provide some mechanism for validation of iterates.

**Form of skill-set**. For the main experiments, we fixed a number of discrete tasks to be learned (without tuning this), but one should consider how the set of skills can be grown or parameterized to have higher capacity (e.g. a multi-label or continuous latent). Otherwise, the task distribution may become overloaded (complicating task inference) or limited in capacity (preventing coverage).

**Accumulation of skill**. We mitigate forgetting with the simple solution of reservoir sampling. Better solutions involve studying an intersection of continual learning and meta-learning.

**Acknowledgments**

We thank the BAIR community for helpful discussion, and Michael Janner and Oleh Rybkin in particular for feedback on an earlier draft. AJ thanks Alexei Efros for his steadfastness and advice, and Sasha Sax and Ashish Kumar for discussion. KH thanks his family for their support. AJ is supported by the PD Soros Fellowship. This work was supported in part by the National Science Foundation, IIS-1651843, IIS-1700697, and IIS-1700696, as well as Google.

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
