[Supplementary Material]

# Appendix A    Derivations

## A.1    Derivation for Trajectory-Level Responsibilities (Section 3.2.1)

Here we show that, assuming independence between states in a trajectory when conditioning on a latent variable, computing the trajectory likelihood as a factorized product of state likelihoods for the E-step in standard EM forces the component responsibilities for all states in the trajectory to be identical. Begin by lower-bounding the log-likelihood of the trajectory dataset with Jensen's inequality:

$$\sum_i \log p(\boldsymbol{\tau}) = \sum_i \log p(\mathbf{s}_1^i, \mathbf{s}_2^i, ..., \mathbf{s}_T^i) \tag{10}$$

$$= \sum_i \log \sum_z p(\mathbf{s}_1^i, \mathbf{s}_2^i, ..., \mathbf{s}_T^i | z) p(z) \tag{11}$$

$$\geq \sum_i \sum_z q_\phi(z | \mathbf{s}_1, \mathbf{s}_2, ..., \mathbf{s}_T) \log \frac{p(\mathbf{s}_1^i, \mathbf{s}_2^i, ..., \mathbf{s}_T^i | z) p(z)}{q_\phi(z | \mathbf{s}_1, \mathbf{s}_2 ... \mathbf{s}_T)} \tag{12}$$

$$= \sum_i \mathbb{E}_{z \sim q_\phi(z | \mathbf{s}_1, \mathbf{s}_2, ..., \mathbf{s}_T)} \log \frac{p(\mathbf{s}_1^i, \mathbf{s}_2^i, ..., \mathbf{s}_T^i | z) p(z)}{q_\phi(z | \mathbf{s}_1, \mathbf{s}_2, ..., \mathbf{s}_T)}. \tag{13}$$

We have introduced the variational distribution $q_\phi(\boldsymbol{\tau}, z)$, where $z$ is a categorical variable. Now, to maximize Eq. 13 with respect to $\phi := (\boldsymbol{\mu}_1, \boldsymbol{\Sigma}_1, \pi_1, ..., \boldsymbol{\mu}_N, \boldsymbol{\Sigma}_N, \pi_N)$, we alternate between an E-step and an M-step, where the E-step is computing

$$q_{ik} = q_\phi(z = k | \mathbf{s}_1^i, \mathbf{s}_2^i, ..., \mathbf{s}_T^i) \tag{14}$$

$$= \frac{q_\phi(\mathbf{s}_1^i, \mathbf{s}_2^i, ..., \mathbf{s}_T^i | z = k) q_\phi(z = k)}{\sum_j q_\phi(\mathbf{s}_1^i, \mathbf{s}_2^i, ..., \mathbf{s}_T^i | z = j) q_\phi(z = j)} \tag{15}$$

$$= \frac{q_\phi(\mathbf{s}_1^i | z = k) q_\phi(\mathbf{s}_2^i | z = k) \cdots q_\phi(\mathbf{s}_T^i | z = k) q_\phi(z = k)}{\sum_j q_\phi(\mathbf{s}_1^i | z = j) q_\phi(\mathbf{s}_2^i | z = j) \cdots q_\phi(\mathbf{s}_T^i | z = j) q_\phi(z = j)}. \tag{16}$$

We assume that each $q_\phi(\mathbf{s} | z = k)$ is Gaussian; the M-step amounts to computing the maximum-likelihood estimate of $\phi$, under the mixture responsibilities from the E-step:

$$\boldsymbol{\mu}_k = \frac{\sum_{i=1}^N \frac{q_{ik}}{T} \sum_{t=1}^T \mathbf{s}_t}{\sum_{i=1}^N q_{ik}} \tag{17}$$

$$\boldsymbol{\Sigma}_k = \frac{\sum_{i=1}^N \frac{q_{ik}}{T} \sum_{t=1}^T (\mathbf{s}_t - \boldsymbol{\mu}_k)(\mathbf{s}_t - \boldsymbol{\mu}_k)^\top}{\sum_{i=1}^N q_{ik}} \tag{18}$$

$$\pi_k = \frac{1}{N} \sum_{i=1}^N q_{ik}. \tag{19}$$

In particular, note that the expressions are independent of $t$. Thus, the posterior $q_\phi(z | \mathbf{s})$ will be, too.

## A.2    CARML M-Step

The objective used to optimize the meta-RL algorithm in the CARML M-step can be interpreted as a sum of cross entropies, resulting in the mutual information plus two additional KL terms:

$$- \mathbb{E}_{\mathbf{s} \sim \pi_\theta(\mathbf{s}|\mathbf{z}), \mathbf{z} \sim q_\phi(\mathbf{z})} \left[ \log q_\phi(\mathbf{s}) - \log q_\phi(\mathbf{s}|\mathbf{z}) \right] \tag{20}$$

$$= - \sum_{\mathbf{z}} q_\phi(\mathbf{z}) \sum_{\mathbf{s}} \pi_\theta(\mathbf{s}|\mathbf{z}) (\log q_\phi(\mathbf{s}) - \log q_\phi(\mathbf{s}|\mathbf{z})) \tag{21}$$

$$= - \sum_{\mathbf{z}} q_\phi(\mathbf{z}) \sum_{\mathbf{s}} \pi(\mathbf{s}|\mathbf{z}) \left( \log \frac{q_\phi(\mathbf{s})}{\pi_\theta(\mathbf{s})} + \log \pi_\theta(\mathbf{s}) - \log \frac{q_\phi(\mathbf{s}|\mathbf{z})}{\pi_\theta(\mathbf{s}|\mathbf{z})} - \log \pi_\theta(\mathbf{s}|\mathbf{z}) \right) \tag{22}$$

$$= H(\pi_\theta(\mathbf{s})) + D_{\mathrm{KL}}(\pi_\theta(\mathbf{s}) \| q_\phi(\mathbf{s})) - H(\pi_\theta(\mathbf{s}|\mathbf{z})) - D_{\mathrm{KL}}(\pi_\theta(\mathbf{s}|\mathbf{z}) \| q_\phi(\mathbf{s}|\mathbf{z})) \tag{23}$$

$$= I(\pi_\theta(\mathbf{s}); q_\phi(\mathbf{z})) + D_{\mathrm{KL}}(\pi_\theta(\mathbf{s}) \| q_\phi(\mathbf{s})) - D_{\mathrm{KL}}(\pi_\theta(\mathbf{s}|\mathbf{z}) \| q_\phi(\mathbf{s}|\mathbf{z})). \tag{24}$$

The first KL term can be interpreted as encouraging exploration with respect to the density of the mixture. The second KL term is the reverse KL term for matching the modes of the mixture.

**Density-based exploration**. In practice, we may want to trade off between exploration and matching the modes of the generative model:

$$r_{\mathbf{z}}(\mathbf{s}) = \lambda \log q_{\phi}(\mathbf{s}|\mathbf{z}) - \log q_{\phi}(\mathbf{s}) \tag{25}$$
$$= (\lambda - 1) \log q_{\phi}(\mathbf{s}|\mathbf{z}) + \log q_{\phi}(\mathbf{z}|\mathbf{s}) - \log q_{\phi}(\mathbf{z}) \tag{26}$$
$$= (\lambda - 1) \log q_{\phi}(\mathbf{s}|\mathbf{z}) + \log q_{\phi}(\mathbf{z}|\mathbf{s}) + C \tag{27}$$

where $C$ is constant with respect to the optimization of $\theta$. Hence, the objective amounts to maximizing discriminability of skills where $\lambda < 1$ yields a bonus for exploring away from the mode of the corresponding skill.

### A.3 Discriminative CARML and DIAYN

Here, we derive a discriminative instantiation of CARML. We begin with the E-step. We leverage the same conditional independence assumption as before, and re-write the trajectory-level MI as the state level MI, assuming that trajectories are all of length $T$:

$$I(\boldsymbol{\tau}; \mathbf{z}) \geq \frac{1}{T} \sum_t I(\mathbf{s}_t; \mathbf{z}) = I(\mathbf{s}; \mathbf{z}) \tag{28}$$

We then decompose MI as the difference between marginal and conditional entropy of the latent, and choose the variational distribution to be the product of a classifier $q_{\phi_c}(\mathbf{z}|\mathbf{s})$ and a density model $q_{\phi_d}(\mathbf{s})$:

$$I(\mathbf{s}; \mathbf{z}) = H(\mathbf{z}) - H(\mathbf{z}|\mathbf{s}) \tag{29}$$
$$= -\sum_{\mathbf{z}} p(\mathbf{z}) \log p(\mathbf{z}) + \sum_{\mathbf{s},\mathbf{z}} \pi_{\theta}(\mathbf{s}, \mathbf{z}) \log \pi_{\theta}(\mathbf{z}|\mathbf{s}) \tag{30}$$
$$\geq -\sum_{\mathbf{z}} p(\mathbf{z}) \log p(\mathbf{z}) + \sum_{\mathbf{s},\mathbf{z}} \pi_{\theta}(\mathbf{s}|\mathbf{z}) p(\mathbf{z}) \log q_{\phi_c}(\mathbf{z}|\mathbf{s}) \tag{31}$$

We fix $\mathbf{z}$ to be a uniformly-distributed categorical variable. The CARML E-step consists of two separate optimizations: supervised learning of $q_{\phi_c}(\mathbf{z}|\mathbf{s})$ with a cross-entropy loss and density estimation of $q_{\phi_d}(\mathbf{s})$:

$$\max_{\phi_c} \mathbb{E}_{\mathbf{z}\sim p(\mathbf{z}), \mathbf{s}\sim\pi_{\theta}(\mathbf{z})} \left[\log q_{\phi_c}(\mathbf{z}|\mathbf{s})\right] \qquad \max_{\phi_d} \mathbb{E}_{\mathbf{z}\sim p(\mathbf{z}), \mathbf{s}\sim\pi_{\theta}(\mathbf{z})} \left[\log q_{\phi_d}(\mathbf{s})\right] \tag{32}$$

For the CARML M-step, we start from the form of the reward in Eq. 26 and manipulate via Bayes':

$$r_{\mathbf{z}}(\mathbf{s}) = \log q_{\phi_c}(\mathbf{z}|\mathbf{s}) + (\lambda - 1) \log q_{\phi_c}(\mathbf{z}|\mathbf{s}) + (\lambda - 1) \log q_{\phi_d}(\mathbf{s}) - (\lambda - 1) \log p(\mathbf{z}) - \log p(\mathbf{z})$$
$$= \lambda \log q_{\phi_c}(\mathbf{z}|\mathbf{s}) + (\lambda - 1) \log q_{\phi_d}(\mathbf{s}) + C \tag{33}$$

where $C$ is constant with respect to the optimization of $\theta$ in the M-step

$$\max_{\theta} \mathbb{E}_{\mathbf{z}\sim q_{\phi}(\mathbf{z}), \mathbf{s}\sim\pi_{\theta}(\mathbf{z})} \left[\lambda \log q_{\phi_c}(\mathbf{z}|\mathbf{s}) + (\lambda - 1) \log q_{\phi_d}(\mathbf{s})\right] \tag{34}$$

To enable a trajectory-level latent $\mathbf{z}$, we want every state in a trajectory to be classified to the same $\mathbf{z}$. This is achievable in a straightforward manner: when training the classifier $q_{\phi_c}(\mathbf{z}|\mathbf{s})$ via supervised learning, label each state in a trajectory with the realization of $\mathbf{z}$ that the policy $\pi_{\theta}(\mathbf{a}|\mathbf{s}, \mathbf{z})$ was conditioned on when generating that trajectory.

**Connection to DIAYN**. Note that with $\lambda = 1$ in Eq. 34, we directly obtain the DIAYN [16] objective without standard policy entropy regularization, and we do away with needing to maintain a density model $\log q_{\phi_d}(\mathbf{s})$, leaving just the discriminator. If $\pi_{\theta}(\mathbf{a}|\mathbf{s}, \mathbf{z})$ is truly a contextual policy (rather than the policy given by adapting a meta-learner), we have recovered the DIAYN algorithm. This allows us to interpret on DIAYN-style algorithms as implicitly doing trajectory-level clustering with a conditional independence assumption between states in a trajectory given the latent. This arises from the weak trajectory-level supervision specified when training the discriminator: all states in a trajectory are assumed to correspond to the same realization of the latent variable.

# Appendix B  Additional Details for Main Experiments

## B.1  CARML Hyperparameters

We train CARML for five iterations, with 500 PPO updates for meta-learning with $RL^2$ in the M-step (i.e. update the mixture model every 500 meta-policy updates). Thus, the CARML unsupervised learning process consumes on the order of 1,000,000 episodes (compared to the ~400,000 episodes needed to train a meta-policy with the true task distribution, as shown in our experiments). We did not heavily tune this number, though we noticed that using too few policy updates (e.g. ~100) before refitting $q_\phi$ resulted in instability insofar as the meta-learner does not adapt to learn the updated task distribution. Each PPO learning update involves sampling 100 tasks with 4 episodes each, for a total of 400 episodes per update. We use 10 PPO epochs per update with a batch size of 100 tasks.

During meta-training, tasks are drawn according to $z \sim q_\phi(z)$, the mixture's latent prior distribution. Unless otherwise stated, we use $\lambda = 0.99$ for all visual meta-RL experiments. For all experiments unless otherwise mentioned, we fix the number of components in our mixture to be $k = 16$. We use a reservoir of 1000 trajectories.

**Temporally Smoothed Reward:** At unsupervised meta-training time, we found it helpful to reward the meta-learner with the average over a small temporal window, i.e. $r_z^W(s_t) = \frac{1}{W} \sum_{i=t-W}^{t} r_z(s_i)$, choosing $W$ to be $W = 10$. This has the effect of smoothing the reward function, thereby regularizing acquired task inference strategies.

**Random Seeds:** The results reported in Figure 6 are averaged across policies (for each treatment) trained with three different random seeds. The performance is averaged across 20 test tasks. The results reported in Figure 7 are based on finetuning CARML policies trained with three different random seeds. We did not observe significant effects of the random seed used in the finetuning procedure of experiments reported for Figure 7.

**Model Selection:** Models used for transfer experiments are selected by performance on a small held-out validation set (ten tasks) for each task, that does not intersect with the test task.

## B.2  Meta-RL with $RL^2$

We adopt the recurrent architecture and hyperparameter settings as specified in the visual maze navigation tasks of Duan et al. [13], except we:

- Use PPO for policy optimization (clip = 0.2, value_coef = 0.1)
- Set the entropy bonus coefficient $\alpha$ in an environment-specific manner. We use $\alpha = 0.001$ for MuJoCo Sawyer and $\alpha = 0.1$ for ViZDoom.
- Enlarge the input observation space to $84 \times 84 \times 3$, adapting the encoder by half the stride in the first convolutional layer.
- Increase the size of the recurrent model (hidden state size 512) and the capacity of the output layer of the RNN (MLP with one hidden layer of dimension 256).
- Allow for four episodes per task (instead of two), since the tasks we consider involve more challenging task inference.
- Use a multi-layer perceptron with one-hidden layer to readout the output for the actor and critic, given the recurrent hidden state.

## B.3  Reward Normalization

A subtle challenge that arises in applying meta-RL across a range of tasks is difference in the statistics of the reward functions encountered, which may affect task inference. Without some form of normalization, the statistics of the rewards of unsupervised meta-training tasks versus those of the downstream tasks may be arbitrarily different, which may interfere with inferring the task. This is especially problematic for $RL^2$ (compared to e.g. MAML [18]), which relies on encoding the reward as a feature at each timestep. We address this issue by whitening the reward at each timestep with running mean and variance computed online, separately for each task from the unsupervised task distribution during meta-training. At test-time, we share these statistics across tasks from the same test task distribution.

Figure 9: Example Observation Sequences from the Sawyer (left) and Vizdoom Random (right) environments.

## B.4 Learning Visual Representations with DeepCluster

To jointly learn visual representations with the mixture model, we adopt the optimization scheme of DeepCluster [10]. The DeepCluster model is parameterized by the weights of a convolutional neural network encoder as well as a $k$-means model in embedding space. It is trained in an EM-like fashion, where the M-step additionally involves training the encoder weights via supervised learning of the image-cluster mapping.

Our contribution is that we employ a modified E-step, as presented in the main text, such that the cluster responsibilities are ensured to be consensual across states in a trajectory in the training data. As shown in our experiments, this allows the model to learn trajectory-level visual representations. The full CARML E-step with DeepCluster is presented below.

---
**Algorithm 3:** CARML E-Step, a Modified EM Procedure, with DeepCluster

---
1: **Require:** a set of trajectories $\mathcal{D} = \{(\mathbf{s}_1, \ldots, \mathbf{s}_T)\}_{i=1}^N$
2: Initialize $\phi := (\phi_w, \phi_m)$, the weights of encoder $g$ and embedding-space mixture model parameters.
3: **while** not converged **do**
4:     Compute $L(\phi_m; \boldsymbol{\tau}, z) = \sum_{\mathbf{s}_t \in \boldsymbol{\tau}} \log q_{\phi_m}(g_{\phi_w}(\mathbf{s}_t)|z)$.
5:     Update via MLE: $\phi_m \leftarrow \arg\max_{\phi'_m} \sum_{i=1}^N L(\phi'_m; \boldsymbol{\tau}_i, z)$.
6:     Obtain training data $\mathcal{D} := \{(\mathbf{s}, y := \arg\max_k q_{\phi_m}(z=k|g_{\phi_w}(\mathbf{s}))\}$.
7:     Update via supervised learning: $\phi_w \leftarrow \arg\max_{\phi'_w} \sum_{(\mathbf{s},y) \in \mathcal{D}} \log q(y|g_{\phi'_w}(\mathbf{s}))$.
8: **Return:** a mixture model $q_\phi(\mathbf{s}, z)$

---

For updating the encoder weights, we use the default hyperparameter settings as described in [10], except 1) we modify the neural network architecture, using a smaller neural network, ResNet-10 [29] with a fixed number of filters (64) for every convolutional layer, and 2) we use number of components $K = 16$, which we did not tune. We tried using a more expressive Gaussian mixture model with full covariances instead of $k$-means (when training the visual representation), but found that this resulted in overfitting. Hence, we use $k$-means until the last iteration of EM, wherein a Gaussian mixture model is fitted under the resulting visual representation.

## B.5 Environments

### B.5.1 ViZDoom Environment

Figure 10: Top-down view of VizDoom environment, with initial agent position. White squares depict stationary objects (only relevant to fixed environment).

The environment used for visual navigation is a 500x500 room built with ViZDoom [32]. We consider both fixed and random environments; for randomly placing objects, the only constraint enforced is that objects should not be within a minimal distance of one another. There are 50 train objects and 50 test objects. The agent's pose is always initialized to be at the top of the room facing forward. We restrict observations from the environment to be $84 \times 84$ RGB images. The maximum episode length is set to 50 timesteps. The hand-crafted reward function corresponds to the inverse $l_2$ distance from the specified target object.

The environment considered is relatively simple in layout, but compared to simple mazes, can provide a more complex observation space insofar as objects are constantly viewed from different poses and in various combinations, and are often occluded. The underlying ground-truth state space is the product of continuous 2D position and continuous pose spaces. There are three discrete actions that correspond to turning right, turning left, and moving forward, allowing translation and rotation in the pose space that can vary based on position; the result is that the effective visitable set of poses is not strictly limited to a subset of the pose space, despite discretized actions.

### B.5.2 Sawyer Environment

Figure 11: Third person view of the Sawyer environment

For visual manipulation, we use a MuJoCo [60] environment involving a simulated Sawyer 7-DOF robotic arm in front of a table, on top of which is an object. The Sawyer arm is controlled by 2D continuous control. It is almost identical to the environment used by prior work such as [40], with the exception that our goal space is that of the object position. The robot pose and object are always initialized to the same position at the top of the room facing forward. We restrict observations from the environment to be $84 \times 84$ RGB images. The maximum episode length is set to 50 timesteps. The hand-crafted reward function corresponds to the negative $l_2$ distance from the specified target object.

## Appendix C    Additional Details for Qualitative Study of $\lambda$

### C.1    Instantiating $q_\phi$ as a VAE

Three factors motivate the use of a variational auto-encoder (VAE) as a generative model for the 2D toy environment. First, a key inductive bias of DeepCluster, namely that randomly initialized convolutional neural networks work surprisingly well, which Caron et al. [10] use to motivate its effectiveness in visual domains, does not apply for our 2D state space. Second, components of a standard Gaussian mixture model are inappropriate for modeling trajectories involving turns. Third, using a VAE allows sampling from a continuous latent, potentially affording an unbounded number of skills.

We construct the VAE model in a manner that enables expressive generative densities $p(\mathbf{s}|z)$ while allowing for computation of the policy reward quantities. We set the VAE latent to be $(\mathbf{z}, t)$, where $p(\mathbf{z}, t) = p(\mathbf{z})p(t) = \mathcal{N}(\mathbf{0}, \boldsymbol{I})\frac{1}{T}$. The form of $p(t)$ follows from restricting the policy to sampling trajectories of length $T$. We factorize the posterior as $q_\phi(\mathbf{z}, t|\mathbf{s}_{t'}) = q(\mathbf{z}|\mathbf{s}_{t'})\delta(t - t')$. Keeping with the idea of having a Markovian reward, we construct the VAE's recognition network such that it takes as input individual states after training. To incorporate the constraint that all states in a trajectory are mapped to the same posterior, we adopt a particular training scheme: we pass in entire trajectories $\mathbf{s}_{1:T}$, and specify the posterior parameters as $\mu_z = \frac{1}{T}\sum_t g_\eta(\mathbf{s}_t)$ and $\sigma_z^2 = \frac{1}{T}\sum_t g_\eta(\mathbf{s}_t)$.

The ELBO for this model is

$$\mathbb{E}_{\mathbf{z}, t \sim q_\phi(\mathbf{z}, t|\mathbf{s}_{t'})}\big[\log q_\phi(\mathbf{s}_{t'}|\mathbf{z}, t)\big] - D_{\text{KL}}(q_\phi(\mathbf{z}, t|\mathbf{s}_{t'}) \parallel p(\mathbf{z}, t)) \tag{35}$$

$$= \mathbb{E}_{\mathbf{z} \sim q_\phi(\mathbf{z}|\mathbf{s}_{t'})}\big[\log q_\phi(\mathbf{s}_{t'}|\mathbf{z}, t')\big] - D_{\text{KL}}(q_\phi(\mathbf{z}|\mathbf{s}_{t'}) \parallel p(\mathbf{z})) - C \tag{36}$$

where $C$ is constant with respect to the learnable parameters. The simplification directly follows from the form of the posterior; we have essentially passed $t'$ through the network unchanged. Notice that the computation of the ELBO for a trajectory leverages the conditional independence in our graphical model.

### C.2 CARML Details

Since we are not interested in meta-transfer for this experiment, we simplify the learning problem to training a contextual policy $\pi_\theta(\mathbf{a}|\mathbf{s}, z)$. To reward the policy using the VAE $q_\phi$, we compute

$$r_z(\mathbf{s}) = \lambda \log q_\phi(\mathbf{s}|z) - \log q_\phi(\mathbf{s}) \tag{37}$$

where

$$\log q_\phi(\mathbf{s}|z) = \log \sum_t q_\phi(\mathbf{s}|z, t)p(t) = \log \frac{1}{T} \sum_t q_\phi(\mathbf{s}|z, t) \tag{38}$$

and we approximate $\log q_\phi(\mathbf{s})$ by its ELBO (Eq. 36), substituting the above expression for the reconstruction term.

## Appendix D   Sawyer Task Distribution

Visualizing the components of the acquired task distribution for the Sawyer domain reveals structure and diversity related to the position of the object as well as the control path taken to effect movement. Red encodes the true position of the object, and light blue that of the end-effector. We find tasks corresponding to moving the object to various locations in the environment, as well as tasks that correspond to moving the arm in a certain way without object interaction. The tasks provide a scaffold for learning to move the object to various regions of the reachable state space.

Since the Sawyer domain is less visually rich than the VizDoom domain, there may be less visually discriminative states that align with semantics of test task distributions. Moreover, since a large part of the observation is proprioceptive, the discriminative clustering representation used for density modeling captures various proprioceptive features that may not involve object interaction. The consequences are two-fold: 1) the gap in the CARML and the object-centric test task distributions may be large, and 2) the CARML tasks may be too diverse in-so-far as tasks share less structure, and inferring each task involves a different control problem.

Figure 12: Skill Maps for Visuomotor Control. Red encodes the true position of the object, and light blue that of the end-effector. The tasks provide a scaffold for learning to move the object to various regions of the reachable state space.

# Appendix E   Mode Collapse in the Task Distribution

Here, we present visualizations of the task distributions induced by variants of the presented method, to illustrate the issue of using an entirely discrimination-based task acquisition approach. Using the fixed VizDoom setting, we compare:

   (i) CARML, the proposed method

  (ii) **online discriminator** – task acquisition with a purely discriminative $q_\phi$ (akin to an online, pixel-observation-based adaptation of [25]);

 (iii) **online pretrained-discriminator** – task acquisition with a discriminative $q_\phi$ as in **(ii)**, initialized with pre-trained observation encoder.

For all discriminative variants, we found it crucial to use a temperature $\geq 3$ to soften the classifier softmax to prevent immediate task mode-collapse.

(i) CARML (ours)        (ii) online discriminator [25]        (iii) pretrained online discriminator

We find the task acquisition of purely discriminative variants **(ii, iii)** to suffer from an effect akin to mode-collapse; the policy's data distribution collapses to a smaller subset of the trajectory space (one or two modes), and tasks correspond to minor variations of these modes. Skill acquisition methods such as DIAYN rely purely on discriminability of states/trajectories under skills, which can be more easily satisfied in high-dimensional observation spaces and can thus lead to such mode-collapse. Moreover, they do not a provide a direct mechanism for furthering exploration once skills are discriminable.

On the other hand, the proposed task acquisition approach (Algorithm 2, §3.2) fits a generative model over jointly learned discriminative features, and is thus not only less susceptible to mode-collapse (w.r.t the policy data distribution), but also allows for density-based exploration (§3.3). Indeed, we find that **(iii)** seems to mitigate mode-collapse – benefiting from a pretrained encoder from **(i)** – but does not entirely prevent it. As shown in the main text (Figure 6c), in terms of meta-transfer to hand-crafted test tasks, the online discriminative variants **(ii, iii)** perform worse than CARML **(i)**, due to lesser diversity in the task distribution.

# Appendix F  Evolution of Task Distribution

Here we consider the evolution of the task distribution in the Random VizDoom environment. The initial tasks (referred to as CARML It. 1) are produced by fitting our deep mixture model to data from a randomly-initialized meta-policy. CARML Its. 2 and 3 correspond to the task distribution after the first and second CARML E-steps, respectively.

We see that the initial tasks tend to be less structured, in so far as the components appear to be noisier and less distinct. With each E-step we see refinement of certain tasks as well as the emergence of others, as the agent's data distribution is shifted by 1) learning the learnable tasks in the current data-distribution, and 2) exploration. In particular, tasks that are "refined" tend to correspond to more simple, exploitative behaviors (i.e. directly heading to an object or a region in the environment, trajectories that are more straight), which may not require exploration to discover. On the other hand, the emergent tasks seem to reflect exploration strategies (i.e. sweeping the space in an efficient manner). We also see the benefit of reorganization that comes from refitting the mixture model, as tasks that were once separate can be combined.

Figure 14: Evolution of the CARML task distribution over 3 iterations of fitting $q_\phi$ in the random ViZDoom visual navigation environment. We observe evidence of task refinement and incorporation of new tasks.