[Reviews · NeurIPS 2019]

Reviewer 1



This work clearly describes their method, as well as contrasts it against a similar approach (DIAYN). In short, they use a data driven method to learn and approximate salient visual features along trajectories as tasks using the DeepCluster method to aggregate new tasks level information. This is additionally important as it avoids GAN style problems that other methods employ, which should, in theory, result in more stable and potentially interpretable results. This work is significant in its simultaneous contributions along unsupervised meta learning as well as vision based clustering in an RL setting, as the authors discuss in section 3.4.

Reviewer 2



Post-rebuttal: Thank you for the responses they have cleared some of the concerns raised. ------- The authors present CARML, an unsupervised method to generate tasks for meta RL. Previous approaches other require the manual definition of task spaces which is especially hard, or by pipelined approaches in which through interactions with a CMP one can yield a task distribution. The authors propose to use a latent variable density model of the meta-learner’s behaviour in order to adapt the task distribution and then meta-learn on those tasks. The paper is well written, everything is motivated and the choices made clearly explained. They provide results of their proposed algorithms in VizDoom and a Sawyer arm, in which they show that their method requires significantly less samples to learn when compared with baseline approaches. I really liked reading the paper, I think unsupervised task generation is very important for meta-learning and this paper provides a nice way of doing it. I do wonder however how D is handled. It seems to me that every trajectory is added to the reservoir. This increases the complexity of fitting the task giver q. Don’t some trajectories become obsolete as the meta-learner evolves? Fitting q in D this way could harm performance. I feel that having a smart way of dealing with the ever increasing D is important. One thing that would have been nice to include is how the approach compares to [22] as it seems to be a motivating paper for this research.

Reviewer 3



This paper presents a method for learning a distribution of tasks to feed to an agent that's learning via meta RL, while simultaneously optimizing the agent to perform better more quickly on tasks sampled from this distribution. The task distribution is trained using an objective that maximizes mutual information between a latent task variable and the trajectories produced by the meta RL agent. The meta RL agent is trained to maximize this mutual information, more or less. The overall optimization relies on some variational lower bounds on mutual information, and on the RL^2 algorithm for meta RL. Experiments are provided which show that the task distributions and meta RL agents trained in this co-adaptive manner exhibit some potentially useful behaviors, e.g. an improved ability to quickly solve new tasks sampled from an "actual" task distribution -- i.e., a task distribution which is not equal to the one that's co-adapted with the agent. I think the ideas explored in this paper are reasonably interesting, and may spark some more practical insights in the rest of the research community. The paper was clear and didn't make unnecessary claims. My main criticism is that it's a bit hard to predict the strength and future value of this work in the absence of prior work that it out-performs or a strong argument for why "automatic curriculum learning for unsupervised meta RL" is a conjunction of keywords worth exploring. Extending that point, there also isn't much effort to explain why this particular approach to this particular problem is particularly worthwhile, which would be helpful in the absence of prior efforts to solve the problem. E.g., as an alternative one could train an agent and task distribution using one of the 10s of DIAYN-like approaches, and then train another agent to solve tasks from this distribution via meta RL using reward functions constructed similarly to the ones used in the proposed method. Post Rebuttal: I appreciate the new experiments aimed at separating the benefits of improved diversity-based skill learning and co-adaptive vs pipelined approaches to incorporating this in a meta RL setting. I'd recommend refactoring the presentation a bit as well, to emphasize that these two contributions -- i.e. improved diverse skill learning compared to DIAYN and a co-adaptive approach to using diverse skill learning for meta RL -- are separate but complementary. It may be difficult for people who aren't already familiar with these topics to perform the appropriate factorization if it is not explicit in the technical presentation.

[Author Response · NeurIPS 2019]

We thank the reviewers for clear and thoughtful feedback, and respond to specific points raised by reviewers below.

**R2:** "how the approach compares to [22]." **R3:** "absence of prior work that it out-performs".

To address the primary concerns of **R2** and **R3**, we present results of new comparisons to Gupta et al. [22] on the Fixed
ViZDoom experimental setting in Table 1. This comparison ([22]) is representative of "train[ing] an agent and task
distribution using one of the 10s of DIAYN-like approaches" (**R3**) before freezing the task distribution and running
meta-learning in a "pipelined" manner. However, we note that [22] considers environments with simpler, ground-truth
state, as opposed to pixel observations.

The compared approaches are: **(i)** [22], which uses DIAYN [13] for task acquisition,
adapted for pixel observations; **(ii)** an ablation of our method – "pipelined CARML"
– more similar to [22], for an apples-to-apples comparison; **(iii)** [22], but initializing
the DIAYN discriminator of with the image encoder of **(ii)**, to address failure modes
of applying [22] in visual domains; and **(iv)** CARML, our full method.

Table 1: Comparing to [22].

| | Avg. Succ. |
|---|---|
| **i.** [22] | 0.291 |
| **ii. Ours**, pipeline | 0.535 |
| **iii.** [22], smart-init | 0.405 |
| **iv. Ours**, full | **0.625** |

**Our approach outperforms [22] on transfer to test tasks.** The benefit of our task acquisition method over that of
DIAYN (which [22] uses) is indicated by the improvement from **(i)** and **(iii)** to **(ii)**. The benefit of using a curriculum
for meta-learning over the pipelined approach of [22] is indicated by the improvement from **(ii)** to **(iv)**. **Please find**
**discussion of these results at the end of the page.** We will include these and further experiments on the remaining
settings in our revision.

**R3**: "Show that ... the newly proposed task is super useful".

We note that the environments considered are nearly identical to the navigation setting of [55] (though ours is more
challenging insofar as no task description is given) and the manipulation setting used in [34], among others. Our
work is among the first to study unsupervised meta-RL in visual domains, addressing challenges of pixel observation
trajectories and partial observability, among others, which exacerbate the challenges of unsupervised RL and meta-RL.

**Populating** $\mathcal{D}$ **(R2).** We choose the simplest strategy that keeps complexity constant: sample a fixed number of
trajectories uniformly at random from the entire history, i.e. reservoir sampling. We used a reservoir of 1000 trajectories
(not tuned). We agree with **R2** that more sophisticated sampling strategies are worth pursuing in future work.

**Comparison Details.** Differing from [22], we use $RL^2$ instead of MAML for more direct comparability; to our
knowledge, policy gradient MAML has yet to be successfully implemented in RL domains with pixel observations.
Comparison **(ii)** uses a contextual policy to co-adapt with the task distribution before freezing the task distribution
and meta-learning with $RL^2$. Results are reported for transfer to the Fixed ViZDoom test tasks, analogous to results
in Figure 5a of submission. We use the same hyper-parameters for skill acquisition (i.e. number of skills) as existing
experiments. In Table 1, we report the average of two runs per approach, but will use more in our revision.

**Comparison Discussion (R2, R3).** We find the task acquisition of DIAYN variants **(i, iii)** to suffer from an effect akin
to mode-collapse; the policy's data distribution collapses to a smaller subset of the trajectory space (one or two modes),
and tasks correspond to minor variations of these modes. Skill acquisition methods such as DIAYN rely purely on
discriminability of states/trajectories under skills, which can be more easily satisfied in high-dimensional observation
spaces and can thus lead to such mode-collapse (related to the instability of GAN methods noted by **R1**). Moreover,
they do not a provide a direct mechanism for furthering exploration once skills are discriminable.

On the other hand, the proposed task acquisition approach (Alg. 2, Sections 3.2, 3.4) **fits a generative model over**
**jointly learned discriminative features**, and is thus not only **less susceptible to mode-collapse** (w.r.t the policy data
distribution), but also allows for density-based exploration (Section 3.3). Indeed, we find that **(iii)** seems to mitigate
mode-collapse – benefiting from a pretrained encoder from **(ii)** – but does not entirely prevent it. Overall, in terms of
meta-transfer to hand-crafted test tasks, the DIAYN variants **(i, iii)** perform worse than pipelined CARML **(ii)**, due to
the poorer diversity in the task distribution. We will incorporate this comparison, as well as additional visualizations
(i.e. skill maps) of all discussed methods, in the revised Appendix.

Moreover, **(ii)** performs worse than "full CARML" **(iv)**. As in the paper, we hypothesize that this is due to the challenge
of meta-learning more complex task distributions – compared to full CARML, the distribution of trajectories eventually
discovered by the contextual policy of **(ii)** may be just as diverse and structured, but meta-learning the corresponding
task distribution directly from scratch is harder. This shows **the benefit of co-adapting tasks with the meta-learner**
**(iv)** as opposed to using a separate agent **(ii)**, and the value of investigating the effects of curricula on meta-learning.

**[13]** Eysenbach, Gupta, Ibarz, Levine. Diversity is all you need: learning skills without a reward function. ICLR 2019.
**[22]** Gupta, Eysenbach, Finn, Levine. Unsupervised meta-learning for reinforcement learning. arXiv:1806.04640, 2018.
**[34]** Nair, Pong, Dalal, Bahl, Lin, and Levine. Visual reinforcement learning with imagined goals. NeurIPS 2018.
**[55]** Chaplot, Sathyendra, Pasumarthi, Rajagopal, Salakhutdinov. Gated-attention architectures for task-oriented
language grounding. AAAI 2018.


[Meta-Review · NeurIPS 2019]

This work makes progress in the unsupervised meta-learning domain with visual features. This work contributes a model for how to automatically learn useful data in an unsupervised sense, and to incorporate that into a meta learner. All three reviewers find the work novel and significant, and hence I recommend acceptance.